

# Cellular production of a counterfeit viral protein confers immunity to infection by a related virus

Benjamin E. Warner[1], Matthew J. Ballinger[1], Pradeep Yerramsetty[1], Jennifer Reed[1], Derek J. Taylor[1], Thomas J. Smith[2] and Jeremy A. Bruenn[1]

[1] Department of Biological Sciences, The State University of New York at Buffalo, Buffalo, NY, USA
[2] Department of Biochemistry and Molecular Biology, University of Texas Medical Branch Galveston, Galveston, TX, United States of America

## ABSTRACT

DNA copies of many non-retroviral RNA virus genes or portions thereof (NIRVs) are present in the nuclear genomes of many eukaryotes. These have often been preserved for millions of years of evolution, suggesting that they play an important cellular function. One possible function is resistance to infection by related viruses. In some cases, this appears to occur through the piRNA system, but in others by way of counterfeit viral proteins encoded by NIRVs. In the fungi, NIRVs may be as long as 1,400 uninterrupted codons. In one such case in the yeast *Debaryomyces hansenii*, one of these genes provides immunity to a related virus by virtue of expression of a counterfeit viral capsid protein, which interferes with assembly of viral capsids by negative complementation. The widespread occurrence of non-retroviral RNA virus genes in eukaryotes may reflect an underappreciated method of host resistance to infection. This work demonstrates for the first time that an endogenous host protein encoded by a gene that has been naturally acquired from a virus and fixed in a eukaryote can interfere with the replication of a related virus and do so by negative complementation.

## INTRODUCTION

The existence of genes derived from non-retroviral RNA viruses within the cellular genomes of eukaryotes (NIRVs) has been puzzling. Originally discovered in insects (*Crochu et al., 2004*), they have now been demonstrated in fungi (*Frank & Wolfe, 2009*; *Liu et al., 2010*; *Taylor & Bruenn, 2009*), plants (*Chiba et al., 2011*), mammals (*Belyi, Levine & Skalka, 2010*; *Horie et al., 2010*; *Taylor et al., 2011*; *Taylor, Leach & Bruenn, 2010*) and other organisms (*Katzourakis & Gifford, 2010*; *Kondo, Chiba & Suzuki, 2015*). The origin of these sequences and their maintenance over millions of years remain to be explained. The origin of NIRVs is probably by adventitious integration via retrotransposon reverse transcription (*Ballinger, Bruenn & Taylor, 2012*), but their maintenance is striking. Although most NIRVs are pseudogenes, many preserve long open reading frames, which in the fungi can be complete genomes reaching more than 1,400 codons in length (*Taylor & Bruenn, 2009*). Some of these sequences have experienced purifying selection (*Ballinger et al., 2013*; *Fort et al.,*

Corresponding author
Jeremy A. Bruenn, cambruen@buffalo.edu

*2012*; *Taylor & Bruenn, 2009*; *Taylor et al., 2011*). There is at least one case in which a viral gene product (in this case a retroviral glycoprotein) has been appropriated for use by the cell (*Weiss & Stoye, 2013*), but the widespread integration and persistence of viral capsid polypeptide genes, RNA dependent RNA polymerase (RdRp) genes, and a variety of other viral genes suggests the construction of a cellular armamentarium directed against RNA viruses.

One mechanism by which NIRVs might serve an antiviral function is through constitutive expression of siRNAs or piRNAs directed against related viruses (*Da Fonseca et al., 2016*; *Honda & Tomonaga, 2016*). This may be the case for flavivirus NIRVs in mosquitos (*Palatini et al., 2017*), but if this were the universal explanation for maintenance of NIRVs it would not explain the maintenance of long open reading frames. In some of the fungi with NIRVs, there is no RNAi system, since argonaut is missing (*Drinnenberg, Fink & Bartel, 2011*; *Drinnenberg et al., 2009*). *Fujino et al. (2014)* demonstrated that a bornavirus-like nucleoprotein element from ground squirrels inhibited the replication of a related bornavirus in human cells (*Fujino et al., 2014*). The ebolavirus vp35 NIRV protein from *Myotis lucifigus* appears to act as an interferon antagonist in ebolavirus infection of human kidney cells (*Kondoh et al., 2017*). The only other known expressed NIRV proteins are coded by totivirus-like genes in fungi (*Taylor & Bruenn, 2009*). We chose one of these fungi, *Debaryomyces hansenii* (which lacks an RNAi system) to test the antiviral function of NIRVs.

*D. hansenii* has NIRVs derived from a totivirus. Closely situated on the same chromosome there is a complete copy of a totivirus genome, including a copy of the capsid polypeptide gene (cp1) (*Taylor & Bruenn, 2009*) and a single copy of a closely related capsid polypeptide gene, cp2 (*Taylor & Bruenn, 2009*). *D. hansenii* belongs to the CTG clade of yeasts, utilizing an alternate genetic code, in which CUG codes for serine rather than leucine, but its totivirus NIRVs are highly similar to the homologous genes in the *Saccharomyces cerevisiae* totivirus ScVL1 (ScVL-A) (*Diamond et al., 1989*; *Fujimura & Wickner, 1988*). We suspected that the function of these NIRVs is to repel invasion by the cognate totivirus, although no totivirus closely related to ScVL1 has been demonstrated in the CTG clade of yeasts. In fact, it has been postulated that genetic code alterations are a means of erecting a barrier against viral infection (*Holmes, 2009*). However, the totiviruses have breached this barrier, since *Scheffersomyces segobiensis*, another member of the CTG clade, is infected by a totivirus related to the *S. cerevisiae* virus ScVLa (ScVL-BC), which has managed to penetrate the codon barrier by losing almost all its CTG codons (*Taylor et al., 2013*). We sought to test one of the *D. hansenii* capsid polypeptide (cp) NIRVs (cp1) for its ability to cure infection with ScVL1, the virus to which it is most closely related.

It has been known for some time that constitutive cellular expression of portions of the totivirus capsid polypeptides will interfere with the viral replication cycle and cure persistent totivirus infections in *S. cerevisiae* (*Yao & Bruenn, 1995*). The capsid of ScVL1, like that of the inner capsids of reoviruses and rotaviruses, is composed of 60 copies of an asymmetric dimer consisting of two copies of cp in two different conformations (*Naitow et al., 2002*). Most of the dimer contacts occur in the first 435 amino acids of the ScVL1 capsid protein (cap), which is 680 amino acids long (*Naitow et al., 2002*). ScVL1 cap N-terminal

**Table 1  Detection of ScV proteins by tandem hybrid MS/MS.** Detection of ScV proteins in PC847 [ScVL1, ScVLa] transformed with pG3 vectors expressing DhV cp1 or its derivative cp1 mutant with a termination codon after the 12th amino acid. The numbers of unique peptides detected for each protein by tandem hybrid MS/MS are indicated.

|  | Cp1 Trypsin | Cp1 Thermolysin | Cp1 mutant Trypsin | Cp1 mutant Thermolysin |
|---|---|---|---|---|
| ScVL1 (L-A) | 0 | 0 | 14 | 17 |
| ScVLa (L-BC) | 18 | 27 | 19 | 23 |

fragments containing at least 475 amino acids are effective in curing ScVL1 (*Yao & Bruenn, 1995*), presumably acting by negative complementation, successfully forming dimers but failing in subsequent viral assembly steps. ScVLa cap N-terminal fragments are also effective in curing ScVLa. However expression of N-terminal fragments of ScVL1 cap does not cure ScVLa or *vice versa*, which is not surprising given that their caps are only 27% identical in sequence. The *D. hansenii* virus NIRV (DhV) cp1 is 42% identical to ScVL1 but only 21% identical to ScVLa, so we expected that it might cure ScVL1 but not ScVLa.

## MATERIALS AND METHODS

Strains. *S. cerevisiae* strains were from Paul Cullen. PC847 is MATa ura3-52 trp1::KAN [ScVL1, ScVLa, ScVM1]. Some isolates lack ScVLa. PC4391 is PPY640 MATa ade2 his2 leu2 trp1 ura3 can1 FUS1::FUR1-lacZ::LEU2 [ScVL1, ScVLa, ScVM1]. *Debaryomyces hansenii* CBS767 was from Jean-Luc Souciet.

Plasmids. pYES2.1 (Thermo Fisher Scientific, Waltham, MA, USA) is a GAL1 expression vector that provides C-terminal VP5 and 6xHis-tags. pG3 is a GAPDH expression vector. Both are *E. coli-S. cerevisiae* shuttle vectors selectable in yeast by URA3 (pYES) or by TRP1 (pG3) and in *E. coli* by ampicillin resistance. Dhvcp1 DNA, optimized for expression in *S. cerevisiae*, was synthesized by BioBasic and cloned into the BamHI site of pG3. It was subsequently subcloned into pYES2.1 using topoisomerase cloning.

RNA preparation. Crude RNA was prepared by whole cell phenol extraction (*Bruenn & Keitz, 1976*). For some experiments (File S1), DNA was removed with RNase free DNase (Promega, Madison, WI, USA) and the DNase denatured by heating.

RTPCR and qRTPCR. RTPCR was performed with the iTaq Universal Probes One-Step Kit (BioRad, Hercules, CA, USA) using the manufacturer's instructions and appropriate specific primers (see below) in a BioRad T100 Thermal Cycler. Cycles were 50 °C for 10 min, 95 °C for 3 min; then 34 cycles of 95 °C for 15 s and 60 °C for 1 min; finally 72 °C for 5 min. qRTPCR was performed with the iTaq™ Universal SYBR®Green Supermix (BioRad, Hercules, CA, USA) in a BioRad C1000 Touch Thermal Cycler using the manufacturer's instructions with appropriate primers (Table 1). Cycles were 2 min at 65 °C, 30 min at 50 °C, 10 min at 95 °C, 30 cycles of (30 s at 94 °C, 30 s at 50 °C, 60 s at 72 °C), 10 min at 72 °C. We analyzed the qPCR data using the $2^{(-\Delta\Delta Ct)}$ method (*Livak & Schmittgen, 2001*) and plotted the results as $\log_{10}$.

Protein purification. Proteins were prepared from whole cells with a proteinase inhibitor cocktail (S8830; Sigma-Aldrich, St. Louis, MO, USA) by French Press followed by low speed centrifugation (5 k × g) and isolation of supernatant. His-tagged proteins were purified from the supernatant with nickel magnetic beads (Biotool, Jupiter, FL, USA) according to the manufacturer's instructions and concentrated for application to SDS-PAGE by acetone precipitation (*Atallah, Flory & Mallick, 2017*). His-tagged proteins were elaborated on 7.5% SDS-PAGE (Mini-PROTEAN TGX gels; BioRad, Hercules, CA, USA).

Western blotting. Proteins were transferred to PVDF filters (Merck-Millipore, Burlington, WA, USA) using Mini-PROTEAN Precast Gels (BioRad) using the manufacturer's instructions and apparatus. Detection of his-tagged proteins was with mouse anti PENTA Histidine Tag:HRP (BioRad, Hercules, CA, USA) and SuperSignal West Dura extended duration substrate (Thermo Fisher Scientific, Waltham, MA, USA) by enhanced chemiluminescence and exposure to CL-XPosure film (Thermo Fisher Scientific). Protein markers were BenchMark His-tagged protein standards (Thermo Fisher Scientific, Waltham, MA, USA) and the PageRuler Plus Prestained Protein Ladder (Thermo Fisher Scientific).

Mass spectroscopy. GeLC-MS/MS was performed on SDS PAGE fractions by the University of Washington Proteomics Facility or by Bioproximity LLC

Structure prediction. Modeling used the ESyPred3D Web Server 1.0 (*Lambert et al., 2002*) or the I-TASSER server (*Roy, Kucukural & Zhang, 2010*). Analysis of hybrid DhV-ScV capsids used the threaded structure of the DhV capsid protein and the structure of the ScVL1(LA) virus (*Naitow et al., 2002*). A portion of the virus capsid was generated using the VIPER database (*Carrillo-Tripp et al., 2009*). Copies of the DhV subunits were then aligned to this LA scaffold using the program COOT (*Emsley & Cowtan, 2004*). Using this assembly, the various interface surfaces were analyzed using the online tool PDBePISA (*Krissinel & Henrick, 2007*).

PCR primers. Primers (IDT) used are described in Table 1.
Note that the expected size for the RPS11B PCR fragment is derived from the mature mRNA, minus the intron in the gene. The DhVcp1 sequence is not that of the original (GenBank accession no. GQ291319.1) but is optimized for expression in *S. cerevisiae* with the same protein sequence as the original (with the one CTG codon corrected to TCG). The ScVL1 sequence is GenBank M28353.1.

## RESULTS

DNAs coding for the DhV cp1 and ScVL1 cap were synthesized. The one CTG codon in cp1 was altered to TCG to preserve the exact protein sequence present in *D. hansenii* and the sequence optimized for codon expression in *S. cerevisiae*. The resultant sequence was cloned into pG3 (*Schena, Picard & Yamamoto, 1991*), a GAPDH expression vector and into pYES2.1, a GAL1 expression vector with several protein tags, including a 6xhis-tag, in the correct orientation for protein expression. Cloning into pYES was performed so that the

6xhis-tag would follow the C-terminus of cp1. Since a portion of the C-terminus of ScVL1 cap is unstructured in the assembled capsid (*Naitow et al., 2002*) and since most of the contacts involved in capsid assembly are in the N-terminus, it was hoped that addition of a protein tag at the C-terminus would not affect assembly.

A derivative strain of PC847 of *S. cerevisae*, in which ScVLa and ScVM1 are absent, was transformed with pYEScp1 and pG3cp1. When the transformants are grown in galactose minimal medium (PC847pYEScp1) or glucose minimal medium (PC847pG3cp1) for many generations, ScVL1 is cured (Fig. 1). The absence of ScVL1 is evident both by gel electrophoresis of dsRNA (top panel) and by RTPCR with primers derived from the RdRp region of ScVL1 (middle panel). Control RTPCR with primers from a ribosomal protein mRNA, bracketing an intron and therefore sensitive only to mature mRNA and not to genomic DNA, are positive with all RNA preparations (bottom panel). Interference with ScVL1 propagation is the result of protein interactions: transformation with similar expression vector constructs in which a single base substitution (T for A) introduces a nonsense codon after codon 12 in the cp1 sequence (pG3cp1m and pYEScp1m) fails to cure ScVL1. That both the tagged (pYEScp1) and untagged (pG3cp1) vector constructs cure ScVL1 suggests that the tagged cp1 is equivalent to the untagged version.

Despite the introduction of an early nonsense codon in cp1, the cp1 mRNA is still fully expressed in the transformants (File S1). Sample data from transformants of *S. cervisiae* strain 4391, which has ScVL1, ScVM1, and ScVLa, are shown. In addition, the specificity for interference with ScVL1 and not ScVLa is evident (Table 2 and File S1). Table 2 shows results from a derivative of PC847 with both ScVL1 and ScVLa. This demonstrates that the cp1 NIRV does cure its cognate virus and does so by expression of a protein, not an RNA. The absence of ScVL1 viral particles in the cp1 transformants but their persistence in the cp1mut transformants was confirmed by tandem hybrid MS/MS of samples from the 70–100 kDa region of a SDS-PAGE of extracted proteins (Table 2). This experiment also demonstrates the curing of ScVL1 but not ScVLa by DhVcp1.

The explanation for this successful totivirus resistance probably resides in the structure of cp1. We have shown previously that DhV cp1 and cp2 are incapable of forming viral particles (*Taylor & Bruenn, 2009*). However, cp1 is predicted to be capable of mixed dimer formation with ScVL1(LA) cap. The 42% sequence identity between the two proteins allows for a quite adequate modeling of cp1 on the known structure of ScVL1(LA) cap (*Naitow et al., 2002*). Modeling using the ESyPred3D Web Server 1.0 (*Lambert et al., 2002*) on ScVL1(LA) cap gives a predicted structure deviating from ScVL1(LA) cap by only 0.30 angstroms RMSD (excellent agreement). Similarly, the I-TASSER server (*Roy, Kucukural & Zhang, 2010*) gives a model with 0.46 angstroms RMSD. Analysis of hybrid DhV-ScV capsids used the threaded structure of the DhV capsid protein and the structure of the ScVL1(LA) virus (*Naitow et al., 2002*). A portion of the virus capsid encompassing the different contacts across icosahedral five-, three-, and two-fold axes was generated using the VIPER database (*Carrillo-Tripp et al., 2009*). Copies of the DhV subunits were then aligned to this LA scaffold using the program COOT (*Emsley & Cowtan, 2004*). Using this assembly, the various interface surfaces were analyzed using the online tool PDBePISA (*Krissinel & Henrick, 2007*) and summarized in the table of Fig. 2. Assembly of virions

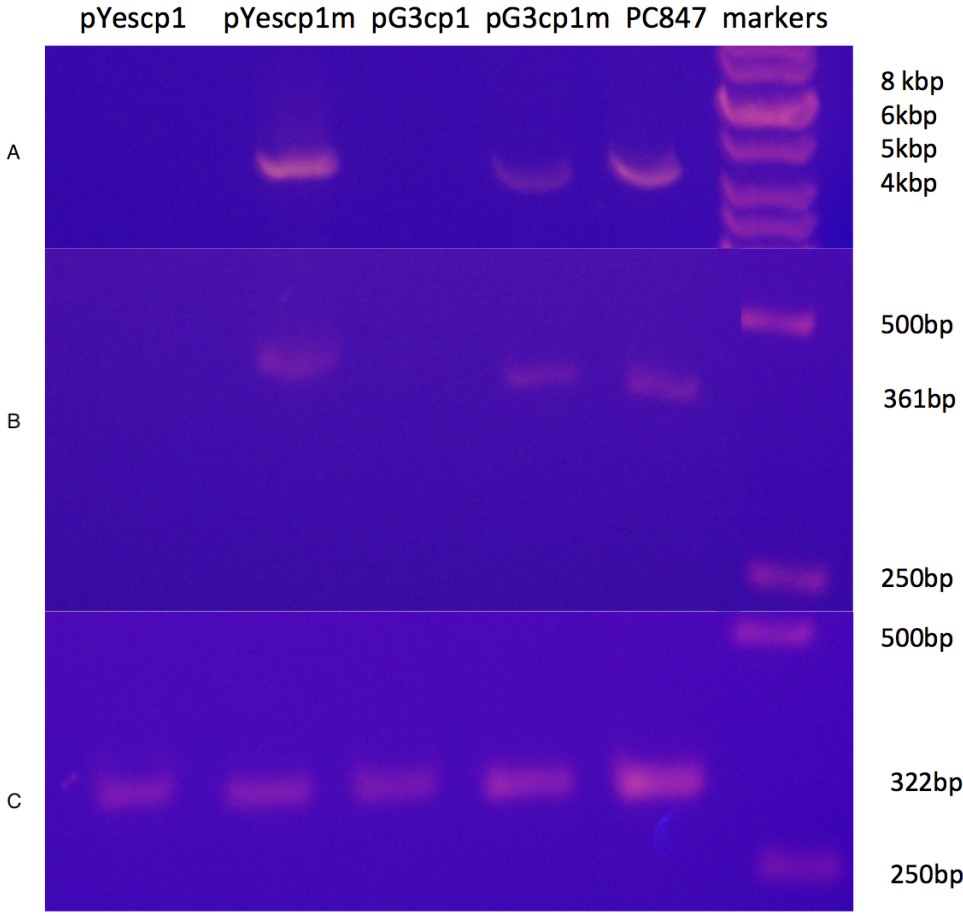

**Figure 1** **Curing of ScVL1(ScVLA) by expression of DhVcp1.** Crude total RNA preparations from pYes transformants grown for many generations in galactose minimal medium and from pG3 transformants grown for many generations in glucose minimal medium were prepared by phenol extraction of whole cells. (A) is crude RNA run on a 1.4% agarose gel. (B) is RTPCR products from these same samples using primers from the RdRp region of L1 on a 1.4% agarose gel. (C) is RTPCR products from the same samples using primers from S. cerevisiae RPS11B, bracketing its intron, again run on a 1.4% agarose gel. Markers were the GeneRuler 1 kb DNA ladder (Thermo Fisher Scientific, Waltham, MA, USA). This is a composite of the three gels run with the same markers.

using these models shows that the subunit interaction interfaces for mixed DhV-ScV subunits are all more stable than the DhV subunits by themselves but all are less stable than the ScV-ScV homo-subunit interfaces (Fig. 2). The interface generating the 3-fold axis of symmetry especially is fatally flawed in the virions with DhV subunits. Consequently, DhVcp1 monomers should participate in particle formation in the presence of ScVL1 cap and cause abortion of particle formation by negative complementation.

This model for interference is corroborated by purification of capsid proteins from PC847pYEScp1. Transformants of PC847 were selected on minimal glucose and then transferred to galactose minimal medium for growth for several generations. If cp1 is capable of forming multimers with ScVL1 cap, mixed multimers should be isolated by

**Table 2  Primers used for RTPCR.**

| Primer | Gene | Sequence | Bases | Product (bp) |
|---|---|---|---|---|
| L1rdrpf | ScVL1 RdRp | CGGCTATATTCGTGTGTGCG | 2,945–2,964 | 361 |
| L1rdrpr | ScVL1 RdRp | TGAGAACATCCTCGCACCTG | 3,305–3,286 | 361 |
| Lardrpf | ScVLa RdRp | AAGAGCACTACCTGACCGTG | 2,731–2,750 | 337 |
| Lardrpr | ScVLa RdRp | CTGCCTCCAGTACTCTTGCC | 3,067–3,048 | 337 |
| L1l | ScVL1 RdRp | TGGAAAAATTTCGGAGAACG | 4,235–4,254 | 230 |
| L1r | ScVL1 RdRp | ATGTTCGCCATTGGTGGTAT | 4,464–4,445 | 230 |
| Lacapl | ScVLa cap | GGCCTGTTATGAATCGAGGA | 1,192–1,211 | 219 |
| Lacapr | ScVLa cap | CAGTTTCCTGCCCCTCAATA | 1,410–1,391 | 219 |
| Cp1l | DhV cp1 | TTTGCCTCCGCAGTAAGTCT | 1,216–1,235 | 206 |
| Cp1r | DhV cp1 | TAAACCAAATACGGCGAACC | 1,421–1,402 | 206 |
| oLR0880 | Actin | GCCTTCTACGTTTCCATCCA | 391–411 | 152 |
| oLR0881 | Actin | GGCCAAATCGATTCTCAAA | 543–523 | 152 |
| RPS11Bf | RPS11B | CCACTGAATTAACTGTTCAATCTG | 5–28 | 322 |
| RPS11Br | RPS11B | ACTGGGACGTTCTTGTGTCT | 837–818 | 322 |

purification using Ni affinity chromatography of his-tagged cp1. Crude protein was isolated from PC847pYEScp1 grown in galactose or glucose minimal medium and PC847pYEScap grown in glucose minimal medium. His-tagged proteins were purified using nickel magnetic beads and elaborated on 7.5% SDS-PAGE. Detection of his-tagged proteins was with an anti-his-tag HRP conjugated antibody and a chemiluminescent substrate. Protein markers were his-tagged protein standards. Both the pYEScp1 and pYEScap transformants produced purified his-tagged proteins of the expected size (Fig. 3, lanes 3 and 4). However, only a minor fraction of cp1 in the crude protein preparation is full-sized (lane 2); most of the cp1 is degraded. Because the his-tag is present on the C-terminus of the protein, only fragments retaining the C-terminus are detectable, but even so, fragments as small as 10 kDa are present. We suspect that degradation is the fate of cp1 monomers not complexed with the viral capsid polypeptide (see 'Discussion'). The control protein from pYEScp1 transformants grown in glucose shows no cp1 at all, as expected (lane 1).

Several regions from an identical gel of purified his-tagged protein from a pYEScp1 PC847 transformant grown for a few generations in galactose were isolated and the included proteins were analyzed by GeLC-MS/MS. The copurification of untagged ScV cap with the his-tagged DhV cp1 implies the formation of mixed multimers. The cp1 his-tagged purified protein is clearly present as a small proportion of cp1-cap multimers; there are about 16.5 cap monomers for every cp1 monomer (Fig. 4). This is consistent with cp1 disrupting cap assembly by negative complementation; the mixed multimers are the result of abortive capsid assembly of the viral capsid from ScVL1 cap and the intrusion of cp1 his tagged monomers. The identification of cp1 and cap was made from numerous tryptic peptides (File S2), essentially the same peptides detected in previous experiments (Taylor et al., 2013) with the addition of one peptide from the C-terminal tag of the cp1 his tagged protein. Note that although the two proteins are 42% identical in sequence, they have no tryptic peptides in common, so they are easily distinguished.

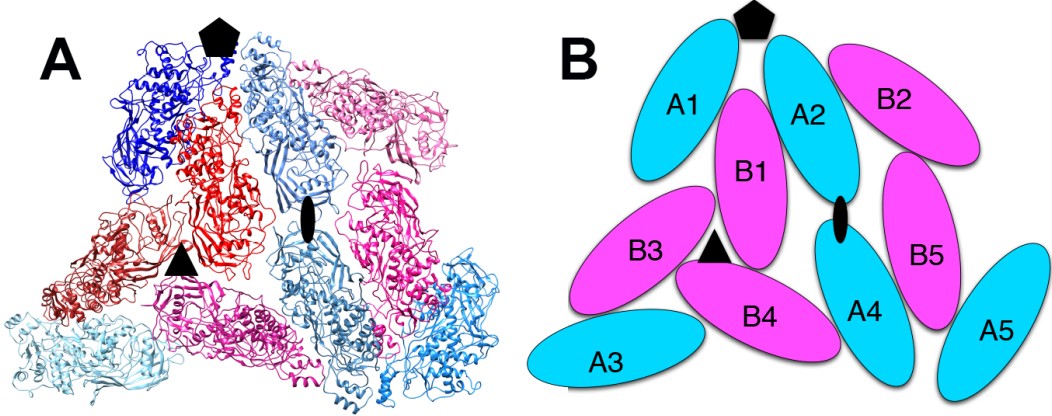

| | LA | | DHV | | A:DHV B:LA | | A:LA B:DHV | |
|---|---|---|---|---|---|---|---|---|
| Contact | Area (Å²) | ΔG | Area (Å²) | ΔG | Area (Å²) | ΔG | Area (Å²) | ΔG |
| A1-B1 | 1384 | -7.9 | 1481 | -2.7 | 1380 | -8.0 | 1452 | -2.9 |
| B1-A2 | 1531 | -7.5 | 1620 | -6.6 | 1588 | -6.5 | 1535 | -8.7 |
| B1-B4 (3-fold) | 583 | -2.6 | 546 | 0.1 | 581 | -2.6 | 546 | 0.1 |
| A2-A4 (2-fold) | 403 | -3.5 | 293 | -1.9 | 296 | -2.0 | 399 | -3.4 |
| A1-B3 | 551 | -4.8 | 609 | -1.6 | 472 | -4.3 | 694 | -1.8 |

**Figure 2** **Predicted stability of DhVcp1-ScVcap interactions.** Energy of formation of subunit surfaces. (A) Ribbon diagram showing two-fold, three-fold, and five-fold axes of symmetry. (B) Subunit labels corresponding to the free energies calculated in (C). (C) Free-energies of formation of subunit interaction interfaces. Note that in the ScVL1(L-A) virus, there are two copies of the capsid protein in the icosahedral asymmetric unit that are not in identical environments, labeled 'A' and 'B' in (B). As shown in the figure, the A subunits are clustered around the five-fold and two-fold axes while the B subunits dominate the three-fold axes.

## DISCUSSION

Clearly, the cp1 NIRV has evolved from a capsid polypeptide capable of virion formation to one that interferes with virion formation. Similar effects have been observed with endogenous retrovirus and retrotransposon capsid genes (*Mura et al., 2004*; *Tucker & Garfinkel, 2016*). Since all dsRNA viruses require an intact capsid for both transcription and replication, this provides a selective advantage to *D. hansenii* by freeing it of its totivirus burden. If totivirus NIRVs provide an antiviral function and are under purifying selection (as evidenced by preservation of extensive open reading frames), persistent totivirus infection must be disadvantageous. At least one activity of totiviruses, stealing of caps from cellular mRNAs (*Fujimura & Esteban, 2011*) and thereby making them untranslatable,

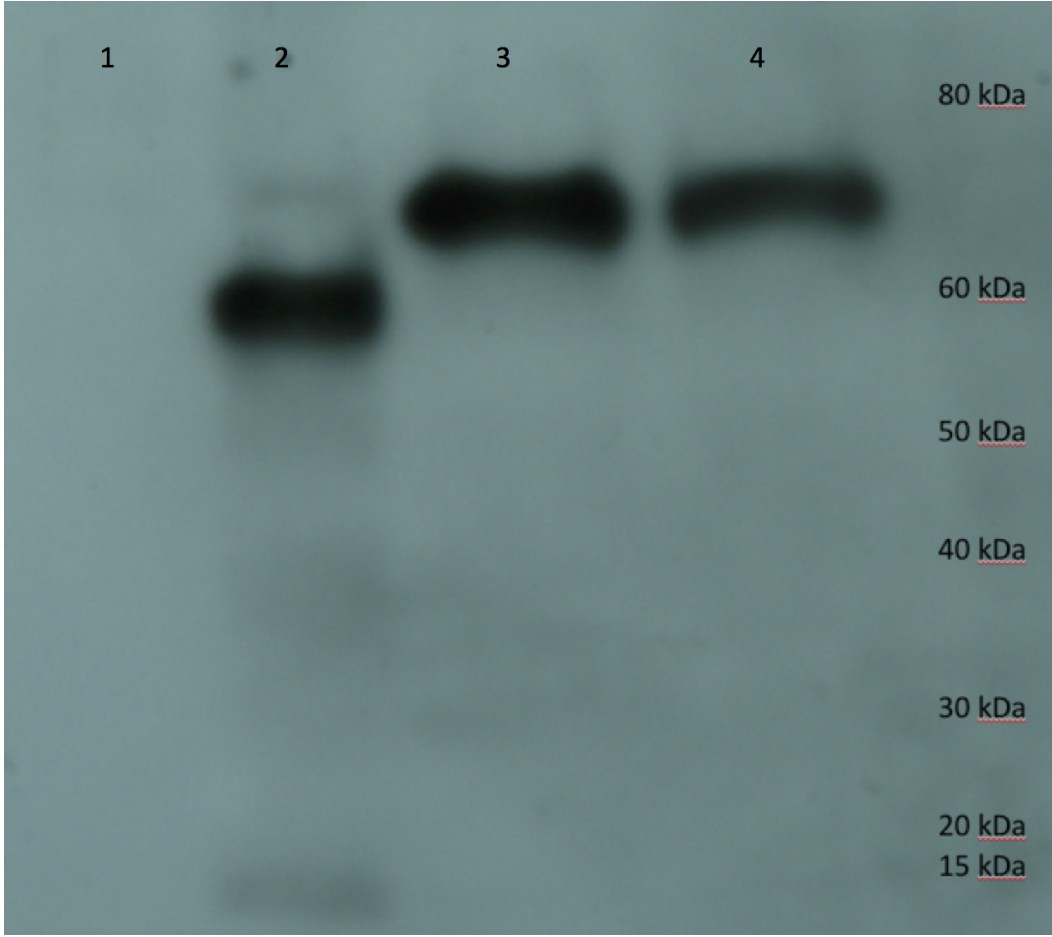

**Figure 3** **Purification of cp1 his-tagged and cap his-tagged proteins.** Western of pYes derived his tagged proteins and elaborated on a 7.5% SDS-PAGE, transferred to PVDF membrane, probed with anti-his tag antibody conjugated to horse radish peroxidase (HRP), developed with an HRP substrate, and exposed to film. Lane 1 is crude protein from pYEScp1 transformed cells grown in glucose; lane 2 is crude protein from pYEScp1 transformed cells grown in galactose; lane 3 is pYEScp1 derived protein (from DhVcp1) purified using nickel affinity magnetic beads; and lane 4 is purified pYEScap1 derived protein (from ScVL1). Markers are his-tagged proteins run in an adjacent lane.

seems likely to be unpalatable to the host. Significantly, most of the predicted proteins of totivirus capsid polypeptide NIRVs have lost this activity (*Taylor & Bruenn, 2009*).

Although the only totivirus presently known in the CTG clade is related to ScVLa(LBC) rather than to ScVL1(LA) (*Taylor et al., 2013*), the persistence of NIRVs related to ScVL1 in *D. hansenii* is not remarkable, given the similar persistence of filovirus NIRVs in mammals not presently exposed to filoviruses (*Taylor, Leach & Bruenn, 2010*). Indeed, one filovirus NIRV does interfere with filovirus infection in susceptible mammalian cells (*Palatini et al., 2017*). Since the *D. hansenii* cp1 protein is rapidly degraded when not present in mixed multimers with ScVL1 capsid protein (Fig. 3), this may explain why it is not detectable in *D. hansenii*, even though its mRNA is present (*Taylor et al., 2013*; *Taylor & Bruenn, 2009*). Presumably, the degraded versions of cp1, even though some retain the his-tag, are not

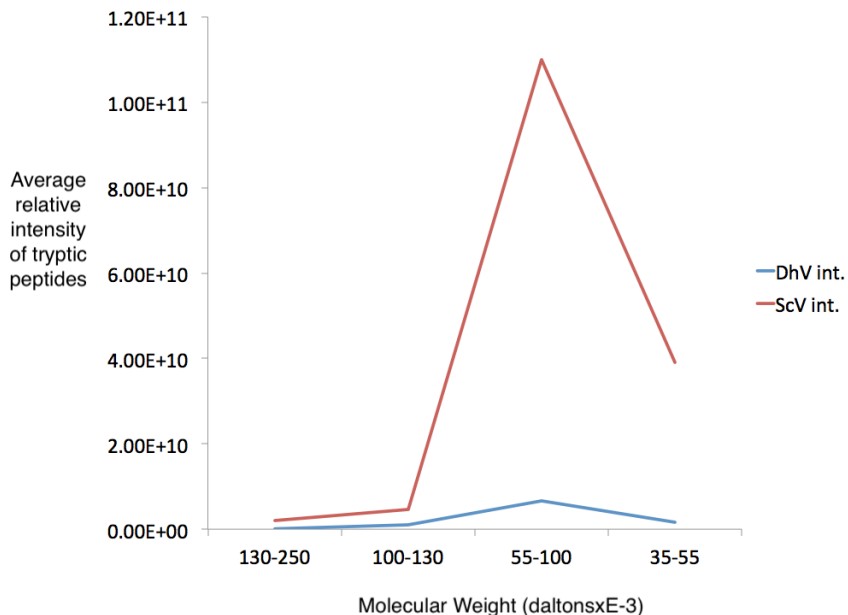

**Figure 4 Detection of cp1 his-tagged—cap multimers.** Relative intensities of pYEScp1 from DhV (DhV int.) and from ScVL1cap (ScV int) from recovered tryptic peptides of gel fragments detected by GelLC-MS/MS from the purified pYEScp1 protein of Fig. 3 from an identical gel run in parallel with the gel of Fig. 3. Four fragments of the gel were analyzed, covering molecular weight ranges of approximately 130–250 kDa, 100–130 kDa, 55–100 kDa, and 35–55 kDa. Both capsid polypeptides were found in the expected molecular weight range, with about 16.5 monomers of ScVcap to each purified his tagged DhVcp1. Intensities were relative molar amounts of the proteins calculated as averages of the relative molar amounts of all detected tryptic peptides.

isolated by affinity chromatography (Fig. 3) because they are folded in such a way that their carboxy termini are not accessible. They are detected by Western blotting because they are completely unfolded in the process of transfer to PVDF membranes.

## CONCLUSION

We conclude that despite a very extensive evolutionary separation of *S. cerevisiae* and *D. hansenii*, the latter preserves a functional NIRV conferring resistance to a virus in the former. Either we have captured a NIRV sequence prior to its disintegration in the absence of its cognate virus or *D. hansenii* continues to retain a functional NIRV because it is periodically exposed to this or a related totivirus by horizontal transfer.

At least in fungi, in which long open reading frames are preserved in NIRVs, their function appears to be production of counterfeit viral proteins altered in such a way that their interaction with elements of the cognate virus disrupts its replication cycle. Production of an authentic NIRV from *D. hansenii* does cure *S. cerevisiae* of its cognate virus, and it does so by negative complementation, rather than by interaction with cellular proteins (*Kondoh et al., 2017*).

## ACKNOWLEDGEMENTS

We thank Yuko Ogata at the University of Washington and Bioproximity, LLC, for mass spectroscopy; BioBasic, Inc. for DNA synthesis; Laura Rusche for help with qRT-PCR experiments; Gerald Koudelka for help in crude protein preparations; and Paul Cullen and Jean-Luc Souciet for strains.

### Funding

Some of this work was supported by NSF grant ARC 1023334 to DJ Taylor. There was no additional funding received for this study. The funders had no role in study design, data collection and analysis, decision to publish, or preparation of the manuscript.

### Grant Disclosures

The following grant information was disclosed by the authors:
NSF grant: ARC 1023334.

### Competing Interests

Jeremy Bruenn is an Academic Editor for PeerJ.

### Author Contributions

- Benjamin E. Warner performed the experiments, authored or reviewed drafts of the paper.
- Matthew J. Ballinger performed the experiments, analyzed the data, contributed reagents/materials/analysis tools, prepared figures and/or tables, authored or reviewed drafts of the paper.
- Pradeep Yerramsetty and Jennifer Reed performed the experiments.
- Derek J. Taylor conceived and designed the experiments, analyzed the data, contributed reagents/materials/analysis tools.
- Thomas J. Smith analyzed the data, contributed reagents/materials/analysis tools.
- Jeremy A. Bruenn conceived and designed the experiments, performed the experiments, analyzed the data, prepared figures and/or tables, authored or reviewed drafts of the paper.

### Data Availability

The raw data are provided in the Supplemental Files.

### Supplemental Information

Supplemental information for this article can be found online at http://dx.doi.org/10.7717/peerj.5679#supplemental-information.

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
