# Peer review of "Cellular production of a counterfeit viral protein confers immunity to infection by a related virus"

_PeerJ, doi:10.7717/peerj.5679_

## Round 0.1 · original submission · Minor Revisions

Please find attached the comments from three independent reviews of your manuscript "Cellular production of a counterfeit viral protein confers immunity to infection by a related virus".

All reviewers thought that the story presented was interesting, but that the evidence for your conclusions could be stronger. They have all suggested additional experiments/analyses to help strengthen the paper, as well as minor corrections to make the text clearer.

Please address all of their comments, paying particular attention to the suggestions of Reviewer 2. If you need more time to complete additional experiments, please let us know.

In addition, I notice that some of the supplementary files have errors:

i) Supp. File 2 - tryptic peptides are in bold, not underlined as stated in the legeng.
ii) rtpcrrps11braw.png is the wrong figure - this is not the bottom panel of Fig 1 but is a repeat of the middle panel.
iii) qrtpcr.pcrd does not open

Please amend these files as necessary.

Best wishes,
Kate Bishop

Reviewer 1 ·

Basic reporting

See below

Experimental design

See below

Validity of the findings

See below

Additional comments

This study demonstrates that exaptation of a non-retroviral RNA virus capsid gene by yeast can provide immunity to the cognate virus. On the whole, the manuscript is clearly written and the results seem unambiguous. However, the manuscript would profit from consideration of
1. precedents of related phenomena provided by retroviruses (JSRV – Mura PNAS 101:11117, 2004; Fv1 – Goff Cell 86:691, 1996) and retrotransposons (Ty1 – Tucker Mobile Genetics Elements 6:e1154639, 2016).
2. How many copies of cp are there in D.hansenii (line 93)? What is cp1 (line 105)?
3. In Fig1 it looks as though pG3cp1m is reduced compared to PC847. Can these data be quantified? How many repeats were performed? How do levels of mRNA compare in mutants with ORFs (note – I couldn’t open SF1)?
4. How do levels of cp1 expressed from pYES compare with those seen in D.hansenii? Can one demonstrate an effect from integrated “endogenous’ cp1 Do the results presented here really go further than those reported in Yao & Bruenn?

Reviewer 2 ·

Basic reporting

no comment

Experimental design

The authors focused one the interaction between introduced viral protein DhVcp1 and viral coat potein ScVcap, and they concluded that the introduction of DhVcp1 interfered the asembling of virus particles, and subsequently affect the persistence of SsVL1 in yeast.
I have some comments for cosideration
1. It will be much fine if authors delete the viral gene that intergraded on the genome of Debaryomyces hansenii, and then carry out transfection test with ScVL-A. If ScVL-A could replicate in the cp1-deleted D. hansenii, this study will be much convince.
2. The expression of cp1 in D. hansenii and in Saccharomyces cerevisiae may not be comparable. At least, authors need provide the expression data.
3. May S. cerevisiae have strong response when a viral gene is introduced in cells? Why did authors use a strain PC847, in which ScVLa is absent? I read the Materias and methods and I was told that PC847 were infected by ScVL1, ScVLa and ScVM1? If authors use the strain which carried both ScVl1, ScVla and satelite virus ScVM1, and perform this study, it will be much convince.

Validity of the findings

This finding is very interesting; it will significantly help us to understand the functions of NIRVs in eucaryotic organisms. But the presented data is shortage of enough convince by my own judgement.

Additional comments

1. The hypothesis in this study is the D. hansenii virus NIRV (DhV) cp1 might cure ScVL1 but not ScVLa due to similarity of the target proteins (coat protein), thus, this study should present the data that DhVcp1 could not cure ScVLa.
2. If authors did not use strains PC4391 and CBS767 in the current study, delete the infomation of these two strains.
3. The dsRNA bands of ScMV1 should appear in Figure 1 (top pane ).

Reviewer 3 ·

Basic reporting

Clear and unambiguous, professional English used throughout: fair

Literature references, sufficient field background/context provided: fine

Professional article structure, figs, tables. Raw data shared: basically fine (see comments to authors)

Self-contained with relevant results to hypotheses: sufficient but additional confirmation is welcomed (see comments to authors)

Experimental design

Experimental design is fine.

Validity of the findings

Findings in the study is important and has great impact. Conclusion and hypothesis was all based on obtained data from in-vivo, in-vitro and in-silico experiments.

Additional comments

The manuscript of Warner et al. reports the first example of contribution of an endogenous non-retroviral RNA virus element (NIRV) on immunity against a yeast RNA virus in the Saccharomyces cerevisiae pathosystem. The authors proposed a hypothesis of mechanism how the NIRV conferred the immunity that an NIRV product originated from probable ancient totiviral capsid protein gene integrated and interfered virion assembly of present virus ScVL1. Although the evidence supporting the hypothesis is not so strong, the finding is very interesting and worth disseminating to researchers as it stands. Followings are minor considerations for improving the paper.

Some fluctuations found: (His-tagged, his tagged) and (6xHis tags, 6xhis tags), ‘His-tagged’ and ‘6xHis-tag’ may be better?
L113: ScVL1 capsid protein (cap) N terminal fragments...
L118: caps means capsids?
L157, L282, L421: Western(s) may be Western blotting
L159: Mouse anti PENTA Histidine Tag:HRP (BioRad)?
L183: the resultant sequence was cloned into...
L203-L210: this paragraph needs more explanation about obtained data; the yeast/virus materials are also deferent from ones in the former paragraph, please describe
L224: give a space after COOT
L242: pYEScap
L251: proteins were analyzed
L255; ScVL1
L233-260: (comments) In my understanding, physical interaction of proteins is usually examined and evidenced by two or three different approaches because of potential false positive output. I guess the authors tried several approaches but suffered from a problem that cp1 expression resulted in degradation as shown in Fig. 3. Yeast two-hybrid system may be useful and easy to confirm the interaction between DhV-cp1 and ScVL1-cap.

Legend to Fig. 1: the upper panel shows electrophoresis of dsRNA fraction
Fig. 1: quality of the figure may need increased especially the middle panel; spaces need between number and kbp/bp; L1 on the right may be ScVL1
Legend to Fig. 4: (ScV int.); figure title and explanation are hard to follow, please restate clearly in a form consistent with description in L249-253
Fig. 4: there is no labeling on X- and Y-axis parameters
Supplementary Fig. 1: this figure carries important data and could be moved to main body of manuscript

---

## Round 0.2 · Minor Revisions

Dear Jeremy,

Thank you for your revisions. I am happy that all the reviewer's comments have either been addressed or that you have provided adequate reasoning for why it is not possible to address them.

There are two minor textual changes to make before I can accept your manuscript:

1) Supp. Fig. 2 legend still doesn't match the figure. Please change the legend to read "tryptic peptides are highlighted in bold"

2) There are too many "of's" in the sentence on line 199 of the track changed manuscript. Please correct this sentence.

Best wishes,
Kate

---

## Round 0.3 · accepted · Accept

Dear Jeremy,

Thanks for your revision. The Figure legend to Sup. Fig. 2 now looks fine.

Unfortunately, the manuscript you resubmitted for version 2 looks the same as the previous version, so the first sentence of the second paragraph of the results section still has a typo (too many "of's" in the sentence: "A derivative of strain of PC847 of S. cerevisiae..."). However, I have accepted the manuscript as I presume you can edit this sentence in the "proofs" stage? I will also flag it up to the PeerJ staff.

Congratulations on your paper!
Best wishes,
Kate

#